# Integrating interferon-gamma release assay testing into provision of tuberculosis preventive therapy is feasible in a tuberculosis high burden resource-limited setting: A mixed methods study

Simon Muchuro[1,2‡], Rita Makabayi-Mugabe[1,2‡]*, Joseph Musaazi[3], Jonathan Mayito[3], Stella Zawedde-Muyanja[1,3], Mabel Nakawooya[2], Didas Tugumisirize[2], Patrick Semanda[4], Steve Wandiga[5], Susan Nabada-Ndidde[4], Abel Nkolo[3], Stavia Turyahabwe[2]

1 USAID-Defeat TB Project, University Research Co. LLC, Kampala, Uganda, 2 Ministry of Health-National TB and Leprosy Division, Kampala, Uganda, 3 The Infectious Diseases Institute, College of Health Sciences, Makerere University, Kampala, Uganda, 4 Ministry of Health-Central Public Health Laboratories, Kampala, Uganda, 5 Kenya Medical Research Institute, Nairobi, Kenya

‡ SM and RMM share co-first authorship on this work.
* rmakabayi@idi.co.ug

## Abstract

The World Health Organization recommends the scale-up of tuberculosis preventive therapy (TPT) for persons at risk of developing active tuberculosis (TB) as a key component to end the global TB epidemic. We sought to determine the feasibility of integrating testing for latent TB infection (LTBI) using interferon-gamma release assays (IGRAs) into the provision of TPT in a resource-limited high TB burden setting. We conducted a parallel convergent mixed methods study at four tertiary referral hospitals. We abstracted details of patients with bacteriologically confirmed pulmonary tuberculosis (PBC TB). We line-listed household contacts (HHCs) of these patients and carried out home visits where we collected demographic data from HHCs, and tested them for both HIV and LTBI. We performed multi-level Poisson regression with robust standard errors to determine the associations between the presence of LTBI and characteristics of HHCs. Qualitative data was collected from health workers and analyzed using inductive thematic analysis. From February to December 2020 we identified 355 HHCs of 86 index TB patients. Among these HHCs, uptake for the IGRA test was 352/355 (99%) while acceptability was 337/352 (95.7%). Of the 352 HHCs that were tested with IGRA, the median age was 18 years (IQR 10–32), 191 (54%) were female and 11 (3%) were HIV positive. A total of 115/352 (32.7%) had a positive IGRA result. Among HHCs who tested negative on IGRA at the initial visit, 146 were retested after 9 months and 5 (3.4%) of these tested positive for LTBI. At multivariable analysis, being aged ≥ 45 years [PR 2.28 (95% CI 1.02, 5.08)], being employed as a casual labourer [PR 1.38 (95% CI 1.19, 1.61)], spending time with the index TB patient every day [PR 2.14 (95% CI 1.51, 3.04)], being a parent/sibling to the index TB patients [PR 1.39 (95% CI 1.21, 1.60)] and sharing the same room with the index TB patients [PR 1.98 (95% CI 1.52, 2.58)] were associated with LTBI.

**Data Availability Statement:** The dataset used and/or analyzed during the current study has been uploaded as part of this submission as supplementary information.

**Funding:** Support for the conduct of this study was provided by the World Bank to the Ministry of Health, Uganda through funding to the East African Public Health Laboratories Networking Project (Project ID: P111556 and Grant recipient: ST) and QIAGEN QIAGEN-Gruppe, Germany. The funders had no role in the study design, data collection, data analysis, decision to publish, or preparation of the manuscript. None of the authors received a salary from the funders.

**Competing interests:** The authors have declared that no competing interests exist.

Implementation challenges included high levels of TB stigma and difficulties in following strict protocols for blood sample storage and transportation. Integrating home-based IGRA testing for LTBI into provision of TB preventive therapy in routine care settings was feasible and resulted in high uptake and acceptability of IGRA tests.

## Background

Tuberculosis (TB) is among the top ten causes of morbidity and mortality. In 2019, an estimated 10.0 million people fell ill with TB, and approximately 1.5 million people died from the disease in the same year [1]. Further, about a quarter of the world (approximately 2 billion persons) is infected with latent TB [2]. Among these, 10–15% will progress to active disease in their lifetime, usually within two years following exposure [1]. The risk of disease progression is increased by certain conditions e.g., age, immunosuppressive states like HIV, diabetes, cancer, and malnutrition [3, 4]. Consequently, the World Health Organization (WHO) outlined provision of TB preventive therapy (TPT) for persons at risk of developing active TB as one of the key components in its strategy to end the global TB epidemic by 2035 [5]. In line with this provision, the WHO updated its guidelines for programmatic management of latent TB infection (LTBI) to recommend TPT for HIV negative household contacts (HHCs) older than 5 years in whom active TB has been ruled out. The guidelines also recommend testing for LTBI using the interferon-gamma release assays (IGRAs), where feasible, to identify individuals who would benefit most from TPT [6].

In Uganda, the WHO symptom screen remains the main stay for ruling out latent TB among persons in close contact with patients with confirmed TB. Although immunological tests e.g., as the Tuberculin Skin Test (TST) have better sensitivity and specificity than the WHO symptom screen, their wide-spread use is limited by the need for cold chain maintenance, inter-reader variability and low specificity due to cross-reactivity with the Bacille Calmette-Guérin (BCG) vaccine and other non-tuberculous mycobacteria. The interferon-gamma release assays (IGRAs) is an alternative immunological test for the presence of LTBI which uses whole blood. This test has several advantages over the TST because its interpretation is not user dependent and the test does not cross react with BCG vaccine resulting in higher specificity [7]. We aimed to explore the feasibility of incorporating LTBI screening using an IGRA test (QuantiFERON-TB Gold Plus test (QFT-Plus) into the national algorithm for management of LTBI among HHCs older than 5 years in Uganda.

## Methods

### Study setting

Between February and December 2020, we conducted a parallel convergent mixed methods study at four tertiary referral hospitals. To get a fair representation of the urban and rural settings, we selected one national referral hospital based in the capital city Kampala (Mulago national referral hospital) and three tertiary referral hospitals (RRH) based in the East (Soroti regional referral hospital), Northwest (Arua regional referral hospital), and West (Hoima regional referral hospital) of the country (Fig 1).

### Sample size determination

The estimated sample size required to estimate prevalence of IGRA positivity among HHCs was 385, assuming prevalence of IGRA positivity of 49%(8), a 95% confidence (standard normal deviate, Z = 1.96) and margin of error of 5%. Inflating the sample size to account for

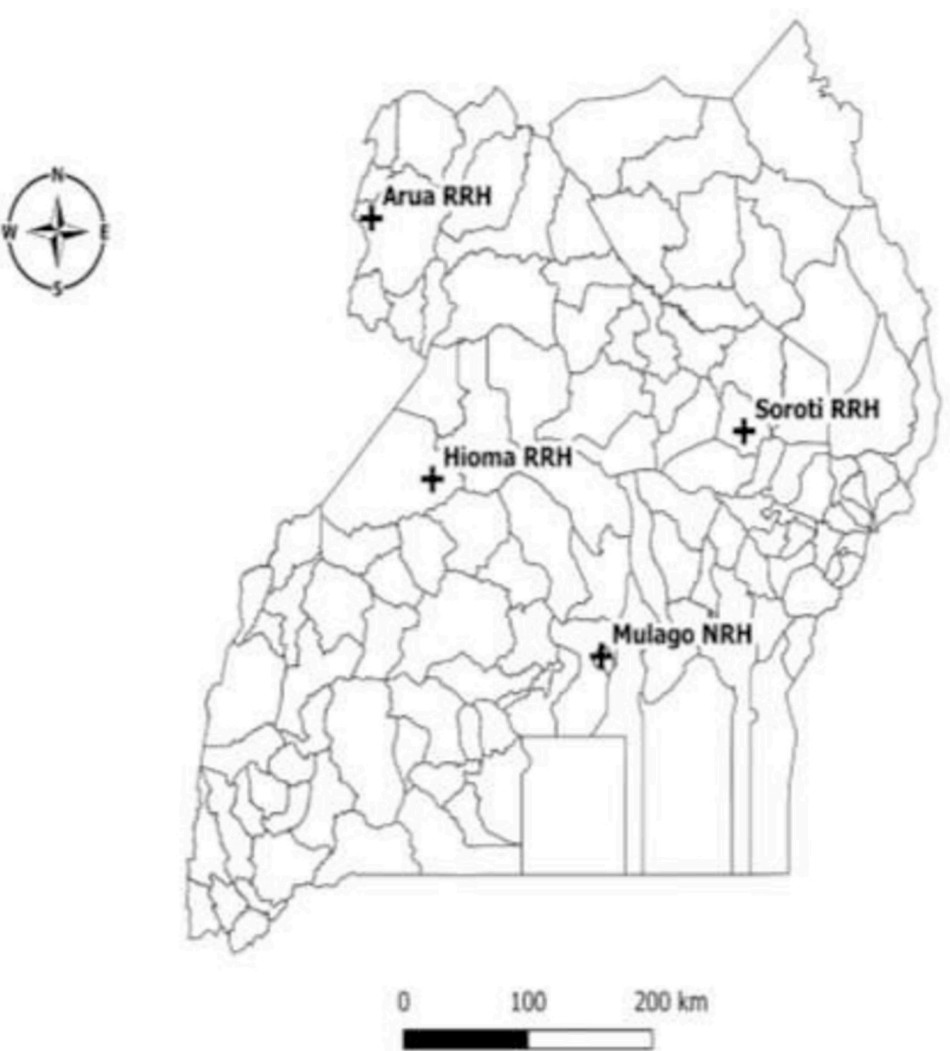

**Fig 1. A map of Uganda showing regional distribution of tertiary referral hospitals during study implementation.**
Link; https://data.humdata.org/dataset/uganda-administrative-boundaries-admin-1-admin-3.

10% of the HHCs that we assumed would have a positive symptom screen for active TB disease, the required sample size was 424 HHCs. We assumed that each index TB patient would have 4 household members [8], and thus based on the sample size of 424, 106 index TB patients were needed to accrue this sample size. However, we attained 352 HHCs from 86 index TB patients due to the limited availability of test kits.

## Data collection

**Selection of index TB patients.** Three months prior to study commencement in February 2020, 619 patients were diagnosed with TB across study sites. Of these, 299 patients were PBCs, of whom 263 were eligible for selection of the 86 required for the study. Those excluded from the study had drug resistant TB or were children under 15 years. HHCs of drug resistant TB patients were not eligible for TPT according to the Ugandan LTBI treatment guidelines at the time of study implementation. However, WHO recommends TPT use in selected high-risk HHCs of patients with drug resistant TB based on individualized risk assessment and clinical

justification [9]. We then used sampling proportionate to size to determine the number of patients to be selected from each hospital. For each hospital, we used systematic random sampling to select the required number of index TB patients. One index TB patient declined study participation due to non-disclosure to a new partner and was replaced with the next consecutive eligible index TB patient at the specific study site. Consequently, 86 index TB patients' homes were visited maintaining sampling proportionate to size for each of the participating hospitals. All HHCs who were eligible for the study and provided informed consent were included in the study.

**Selection of household contacts.**   Data collection among HHCs was carried out between February and December 2020 after obtaining permission from index TB patients to visit their homes to carry out household contact tracing. The study team consisted of qualified health workers who underwent 4 days training on the study protocol and procedures prior to implementation. A team comprising of a clinician (either a nurse, clinical officer, or doctor), counselor, and laboratory technician/phlebotomist conversant with the local dialect visited the index TB patient's home on a scheduled day and requested HHCs to consent to participate in the study.

Detailed information about the study was provided and consent or ascent for screening and enrolment into the study was sought. The study team line listed all HHCs who consented to study participation excluding those who were <5 years, with a history of TPT within the past two years or currently on TB treatment. We screened HHCs using the WHO symptom screen. This involved asking the study participants if they had cough of any duration, weight loss, fevers, and night sweats. For all HHCs without signs and symptoms of TB, we collected socio-demographic data, home-based blood sample collection for LTBI testing using QFT-Plus (*manufactured by QIAGEN QIAGEN-Gruppe Germany*) and HIV counselling and testing (if HIV status was reported as negative or unknown). Study team phlebotomist collected five milliliters (mls) of whole blood: four mls for the IGRA test and 1 ml for HIV 1 & 2 testing using the national testing algorithm. Blood samples collected from the capital city (Kampala) were transported in QFT-plus blood-collection-tubes within the recommended 16 hours to the central laboratory, while blood samples collected from distant study sites were kept at room temperature for utmost three hours and transported in lithium heparin tubes in ice-cold boxes maintained at 2–8 $^{0}$C within 48hours to the central public health laboratory. In addition, we collected information on duration and nature of contact with the index patient. Data was collected electronically using the open data toolkit (ODK).

All asymptomatic HHCs who tested positive on IGRA test were initiated on six months of isoniazid preventive therapy (IPT) while those who tested negative on the initial IGRA test had a second home-based IGRA test performed after nine months. The repeat IGRA test was initially planned to be done at 6 months to rule out LTBI. As a result of Covid-19 travel restrictions, it was performed at 9 months. Those found to be positive on the second test were initiated on TPT.

**Qualitative data.**   Qualitative data was collected from four tertiary hospitals through focus group discussions (FGDs) and key informant interviews (KIIs). Two focus FDGs were held for the participants from Mulago hospital because they had large teams that would meet the criteria for holding an FGD. KII were conducted across other RRHs. The days for the FDGs were specially arranged and the participants were informed on the agenda, date, and approximate duration of the meeting prior to the meeting. The FGDs were conducted in English. All discussions were audiotaped and transcribed. Participant identifiers were not used, but individual participants provided written informed consent and were assigned codes, e.g., five group members will be assigned 01–05. Individual responses in each group were coded by item. Using a phenomenological approach, we explored the experiences of health workers focusing

on their experiences during IGRA study implementation. We purposively sampled health workers who had been involved in contact tracing & implementation of IGRA. Sampling was based on purposeful maximum variation that involved distinct categories of participants like nurses, clinicians, and laboratory technicians. The majority were laboratory staff this being a predominantly lab-based test, involving home-based blood draws, packaging, and transportation of blood samples to the central laboratory. Similarly, both females and males were included in the study. The interview guide consisted of five open ended questions with probes (S1 Text) and follow up questions to create additional depth. Interview questions were developed based on additional information required, the questions were kept sufficiently broad to encourage new concepts to emerge and minimize interviewer bias. Data collection and analysis was led by an independent senior behavioural scientist (AT) who was assisted by members of the research team. We interviewed respondents until saturation was achieved.

*Study definitions.* For this study, we defined a bacteriologically confirmed TB patient as one with a positive Xpert MTB Rif test or positive sputum smear [10], an index case of TB as the initially identified case of new or recurrent TB in a person of any age with bacteriologically confirmed TB diagnosis, and a HHC as a person who shared the same enclosed living space as the index case for one or more nights or for frequent or extended daytime periods during the three months before the start of current treatment [6]. Finally, we defined LTBI as the presence of a positive IGRA test either on the date of first testing or on the date of second testing nine months later.

## Data analysis

**Quantitative data.** We analyzed the data in Stata version 16.1 Special Edition (StataCorp, College Station, Texas, USA). We summarized the characteristics of study participants using frequencies and percentages for categorical variables, and medians with interquartile ranges for continuous variables like age. Study outcomes: IGRA test uptake, acceptability, and IGRA test positivity was summarized as frequencies, proportions, and compared across participants' characteristics using Chi-square test or Fisher's Exact if expected counts are less than 5. IGRA uptake was determined as the proportion of household contacts who took the IGRA test out of all contacts screened and were eligible to take the test. A multivariable multi-level Poisson regression model with exchangeable covariance matrix was used to examine factors associated with LTBI. Robust standard errors were used to correct for overdispersion. Variables were entered into the multivariable regression analysis if they had a p-value of <0.2 at unadjusted analysis. We used variance inflation factors (VIFs) to evaluate multicollinearity in fitted models, where in VIFs >10 were indicative of severe multicollinearity. Analyses were not corrected for multiplicity given the exploratory nature of the study.

**Qualitative data.** Qualitative interviews were coded using an inductive approach with descriptive thematic coding. Interview transcripts, recordings and notes were reviewed for content related to the research question and a coding frame developed with flexibility to accommodate emergent new themes as coding evolved. Using the framework, each transcript was read and reread for recurrent ideas. Codes were assigned to relevant segments of the text; similar codes were aggregated to form themes that were then used to address the research questions and develop coherent narratives [11]. The initial coding framework was developed by a senior behavioral scientist (AT) experienced in qualitative research after reviewing 5% of the transcripts. Subsequent analyses of transcripts were carried out by two members of the research team (RMM and SM) who then compared and discussed their findings. Discrepancies were resolved by mutual agreement. To ensure trustworthiness, transcripts were coded independently, compared, discussed [12].

**Ethics statement.** The study protocol was approved by the Mengo Hospital Research & Ethics Committee (MHREC 57/5–2019) and the Uganda National Council of Science and Technology (UNCST HS 2721). All HHCs provided written informed consent and assent (for participants younger than 18 years) before undergoing any study related procedures. Similarly, written informed consent, including consent to audio-record interviews was obtained from healthcare workers who participated in the qualitative interviews.

## Results

Between February and December 2020, we visited 86 households of index TB patients and identified 355 HHCs, of whom 352 (99.2%) accepted IGRA test. The median number of contacts per index TB patient were six and inter-quartile range of three and seven contacts. The proportion of indeterminate IGRA test results were 1% and 11% at baseline and at repeat testing on follow-up respectively. Fig 2 below shows the flow of study participants through the study.

Of the 352 HHCs on whom IGRA test was done, 54% were female with a median age of 18 years (IQR 10–32), 61% had no employment of whom 64% (138/214) were children of school going age (5 to 14 years), the majority had at-least attained primary level education (>80%), while 73% were HIV negative (Table 1).

## Uptake and acceptability of IGRA test

IGRA test uptake was 99.2% (352/355) (Fig 2). Of 352 that offered a blood sample for the IGRA test, 95.7% said their phlebotomy experience was good or excellent. The 4.3% that reported a bad phlebotomy experience were mainly among the younger age group, notably

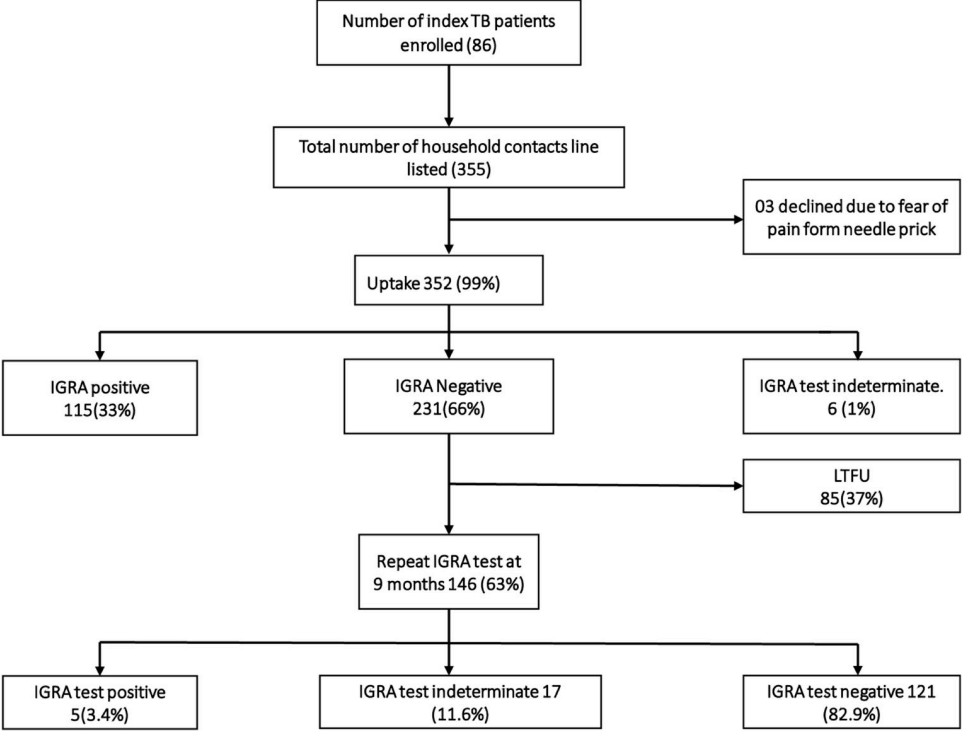

**Fig 2. A consort diagram showing participants' flow during study implementation.**

**Table 1. Participants' characteristics.**

| Characteristics | Number (%), N = 352 |
|---|---|
| *Gender* | |
| Female | 191(54.3) |
| Male | 161(45.7) |
| Age in years, median (IQR) | 18(10–32) |
| *Age groups* | |
| 5–14 | 138(39.2) |
| 15–24 | 84(23.9) |
| 25–44 | 91(25.8) |
| $\geq$ 45 | 39(11.1) |
| *Educational level* | |
| None | 50(14.2) |
| Primary | 206(58.5) |
| Post primary | 96(27.3) |
| *Employment type¶* | |
| None | 46(25.0) |
| Formal employment | 15(8.1) |
| Business | 50(.27.2) |
| Casual laborer | 21(11.4) |
| Agriculture | 52(28.3) |
| *Smoking status* | |
| Never | 328(93.2) |
| Ex-smoker | 5(1.4) |
| Current smoker | 19(5.4) |
| Live with a smoker (Yes) | 121(34.4) |
| *HIV status* | |
| Negative | 257(73.0) |
| Positive | 11(3.1) |
| Unknown* | 84(23.9) |

¶ % computed out of 184 participants after excluding 168 participants under none employment who were aged<18 years.

* Includes those either who did not consent to the test or unavailability of HIV test kits at some sites at the time of the study.

due to pain. Older age (P-value <0.01), level of education (P-value = 0.02) and health facility (P-value <0.01) were significantly associated with acceptability of IGRA test among HHCs (Table 2, P values unadjusted)

## Prevalence of latent TB infection

Of 352 household contacts on whom IGRA test was done, 115 (32.7%) had LTBI on the first IGRA test. Among the 231 who did not have LTBI, 146 (63.2%) received repeat IGRA testing at nine months, of whom 5(3.4%) had LTBI. Therefore, the total number of HHCs with LTBI in this study was 120/352 (34.1%) (Fig 1).

## Factors associated with a positive IGRA test

At multivariable analysis, being aged $\geq$ 45 years compared to age 5–14 years [Prevalence ratio (PR) 2.28 (95% CI 1.02, 5.08)]; being employed as a causal labourer compared to no

**Table 2. Acceptability of IGRA test.**

| Factor | IGRA acceptability level | | | |
|---|---|---|---|---|
| | *Poor n (%)* | *Good or Excellent n (%)* | *Total participants (N)* | *P-value** |
| Overall | 15 (4.3) | 337 (95.7) | 352 | |
| Gender | | | | |
| Female | 8 (4.2) | 183 (95.8) | 191 | 0.94 |
| Male | 7 (4.3) | 154 (95.7) | 161 | |
| *Age group in years* | | | | |
| 5–14 | 14 (10.1) | 124 (89.9) | 138 | <0.01 |
| 15–24 | 1 (1.2) | 83 (98.8) | 84 | |
| 25–44 | 0 | 91 (100.0) | 91 | |
| ≥ 45 | 0 | 39 (100.0) | 39 | |
| *Educational level* | | | | |
| None | 0 | 50 (100) | 50 | 0.02 |
| Primary | 14 (6.8) | 192 (93.2) | 206 | |
| Post primary | 1 (1.0) | 95 (99.0) | 96 | |
| *Health facility* | | | | |
| Mulago NRH | 0 | 140 (100.0) | 140 | <0.01** |
| Hoima RRH | 15 (21.1) | 56 (78.9) | 71 | |
| Soroti RRH | 0 | 71 (100.0) | 71 | |
| Arua RRH | 0 | 70 (100.0) | 70 | |

* P-value from Chi-square

**P-value from Fisher's exact test

employment [PR 1.38 (95% CI 1.19, 1.61)]; spending time with the index TB patient everyday compared to not every day [PR 2.14 (95% CI 1.51, 3.04)]; sleeping in the same room with the index TB patient compared to sleeping in different houses [PR 1.98 (95% CI 1.52, 2.58)]; and being a parent/sibling to the index TB patients compared other relationship with index [PR 1.39 (95% CI 1.21, 1.60)] were significantly associated with having LTBI (Table 3). No severe multicollinearity was noted, all VIFs were below 10.

## Index TB patients' information

Information from 53 out of 86 index TB patients was accessed at the health facilities, of whom, majority were male (73.6%) with a median age of 32, 98% were new cases of bacteriologically confirmed pulmonary tuberculosis (Table 4).

## Qualitative results

**Characteristics of the qualitative arm participants.** In March 2020, we carried out two focus group discussions (FGDs) each with five participants, and 14 key informant interviews (KII) giving a total of 24 healthcare worker participants in this study. Thirteen of these (54%) were male. There were seven laboratory technicians, five nurses, two counsellors, three community healthcare workers, two clinical officers, two doctors, one laboratory scientist, one quantitative economist and one physician.

Several key themes emerged from the data regarding the health workers experiences, challenges, and barriers to implementation of LTBI screening using IGRA.

**Positive health worker experience during implementation of LTBI screening using IGRA.** Multi-disciplinary teams coupled with the eagerness and self-motivation of the health

**Table 3. Factors associated with latent TB infection among household contacts of patients with bacteriologically confirmed TB.**

| Factor | Number | IGRA positive, n(%) † | Unadjusted | | Adjusted ‡ | |
|---|---|---|---|---|---|---|
| | | | PR (95%CI) | P-value | PR (95%CI) | P-value |
| Overall | 352 | 115 (32.7) | N/A | N/A | | |
| Site location | | | | | | |
| Rural | 212 | 61 (28.8) | 1 | | 1 | |
| Urban | 140 | 54 (38.6) | 1.27 (1.08, 1.49) | <0.01 | 1.04 (0.88, 1.23) | 0.66 |
| Gender | | | | | | |
| Female | 191 | 64 (33.5) | Reference | | | |
| Male | 161 | 51 (31.7) | 1.00 (0.96, 1.05) | 0.92 | - | - |
| Age groups | | | | | | |
| 5–14 | 138 | 28 (20.3) | Reference | | Reference | |
| 15–24 | 84 | 24 (28.6) | 1.14 (0.86, 1.50) | 0.36 | 1.05 (0.73, 1.49) | 0.80 |
| 25–44 | 91 | 38 (41.8) | 1.76 (0.99, 3.13) | 0.05 | 1.38 (0.65, 2.92) | 0.40 |
| $\geq$ 45 | 39 | 25 (64.1) | 2.56 (1.47, 4.46) | <0.01 | 2.28 (1.02, 5.08) | 0.04 |
| Educational level | | | | | | |
| None | 50 | 17 (34.0) | Reference | | | |
| Primary | 206 | 60 (29.1) | 0.90 (0.77, 1.07) | 0.23 | - | - |
| Post primary | 96 | 38 (39.6) | 1.10 (0.70, 1.72) | 0.69 | - | - |
| Employment type | | | | | | |
| None | 214 | 53 (24.8) | Reference | | Reference | |
| Formal employment | 15 | 7 (46.7) | 1.61 (0.87, 2.95) | 0.13 | 1.33 (0.92, 1.91) | 0.13 |
| Business | 50 | 21 (42.0) | 1.54 (1.13, 2.10) | 0.01 | 1.21 (0.99, 1.46) | 0.05 |
| Casual laborer | 21 | 11 (52.4) | 1.81 (1.19, 2.74) | 0.01 | 1.38 (1.19, 1.61) | <0.01 |
| Peasant farming | 52 | 23 (44.2) | 1.66 (1.46, 1.88) | <0.01 | 1.19 (0.78, 1.82) | 0.42 |
| Smoking status | | | | | | |
| Never | 328 | 105(32.0) | Reference | | | |
| Ex-smoker | 5 | 3(60.0) | 1.68 (0.97, 2.90) | 0.06 | - | - |
| Current smoker | 19 | 7(36.8) | 1.01 (0.62, 1.65) | 0.97 | - | - |
| Live with a smoker | | | | | | |
| No | 231 | 73(31.6) | Reference | | | |
| Yes | 121 | 42(34.7) | 1.05 (0.74, 1.48) | 0.79 | - | - |
| HIV status | | | | | | |
| Negative | 257 | 85(33.1) | Reference | | | |
| Positive | 11 | 5(45.4) | 1.23 (0.88, 1.74) | 0.21 | - | - |
| Unknown | 84 | 25(29.8) | 0.97 (0.82, 1.16) | 0.77 | - | - |
| BCG scar present | | | | | | |
| No | 35 | 14(40.0) | Reference | | | |
| Yes | 317 | 101(31.9) | 0.85 (0.68, 1.06) | 0.15 | - | - |
| Time spent with TB index | | | | | | |
| Not every day | 15 | 2 (13.3) | Reference | | Reference | |
| Everyday | 337 | 113 (33.5) | 2.84 (1.74, 4.66) | <0.01 | 2.14 (1.51, 3.04) | <0.01 |
| Contact proximity with TB index Sleeps in | | | | | | |
| [#]Different house | 197 | 46(23.3) | Reference | | Reference | |
| Same house / different room | 86 | 28(32.6) | 1.27 (0.94, 1.71) | 0.12 | 1.11 (0.76, 1.61) | 0.60 |
| Same room | 69 | 41(59.4) | 2.26 (1.78, 2.86) | <0.01 | 1.98 (1.52, 2.58) | <0.01 |
| Relationship with index TB case | | | | | | |
| Others | 112 | 24(21.4) | Reference | | Reference | |
| Parent/Sibling | 221 | 78(35.3) | 1.53 (1.16, 2.00) | <0.01 | 1.39 (1.21, 1.60) | <0.01 |

*(Continued)*

**Table 3.** (Continued)

| Factor | Number | IGRA positive, n(%) † | Unadjusted | | Adjusted ‡ | |
| --- | --- | --- | --- | --- | --- | --- |
| | | | PR (95%CI) | P-value | PR (95%CI) | P-value |
| Spouse | 19 | 13 (68.4) | 2.68 (1.57, 4.56) | <0.01 | 1.28 (0.74, 2.23) | 0.38 |

PR–Prevalence ratio; obtained from multi-level Poisson regression model with robust standard errors.

N–Total number of household contacts of index TB patient

N/A–Not Applicable

† Percent of QFT plus positives of the total within factor category

¶ Missing values: Bacillary load on Xpert (n = 22)

‡ analysis performed on all the 352 participants; Bacillary load which had missing data was excluded due to large P-value (>0.3 at unadjusted, the prespecified threshold for inclusion into adjusted analysis)

#Different house- a homestead setting typically found in rural settings with adjacent outbuildings.

workers to find out the burden of LTBI among HHCs of index TB patients enabled the smooth implementation of IGRA testing in the community.

> *"My motivation was the fact that I work on the TB ward, I also wanted to see how latent TB infection is common. . .. I wanted to know the prevalence of contacts having the infection, so I was interested in knowing that." (KII_Nursing Officer_Hoima_12)*

> *"I expected low numbers in communities to have IGRA positive results. . . but then I got to know that latent TB is real. . ." (FGD_IGRA study team_Mulago_2 _NRH_2)*

**Table 4.** TB index characteristics information.

| | Number (%) N = 53* |
| --- | --- |
| **Sex** | |
| Female | 14(26.4) |
| Male | 39(73.6) |
| **Age** | |
| Median age in years (interquartile range) | 32 (26–46) |
| Age group (years) | |
| 15–17 | 3(5.6) |
| 18–24 | 9(17.0) |
| 25–34 | 17(32.1) |
| 35–44 | 10(18.9) |
| ≥45 | 14(26.4) |
| **TB disease classification** | |
| Bacteriologically confirmed pulmonary TB | 53(98.1) |
| **TB patient type** | |
| New | 46(86.8) |
| Relapse | 4(7.6) |
| Re-treatment after failure | 3(5.7) |
| **HIV status** | |
| Negative | 47(88.7) |
| Positive | 6(11.3) |

*Information on only 53/86 index TB patients could be accessed from the participating hospitals. Information for 33/86 index TB patients could not be accessed due to site related challenges.

*Importance of IGRA and its usefulness versus symptom screen as the current standard of care.* The healthcare workers said that LTBI screening using IGRA helped them better appreciate the importance of TB preventive therapy. The exercise also helped them realize the importance of testing before treating for LTBI so as to target the limited supplies of TPT to those who need it most and lessen the chances of toxicities.

*"...and because we know, if someone is positive for latent TB, there are chances that he can progress to active TB. So..., those that are positive are given some therapy. (KII_Labtechnologist_1_CPHL_10)*

*"...if we continue with the current standard... we expose people who do not truly have latent tuberculosis to a treatment that—1) is not going to benefit them, and 2) is going to expose them to toxicity... (FGD_IGRA study team_Mulago_2 _NRH_2)*

**Barriers to implementation of LTBI screening using IGRA.** The healthcare workers reported some challenges with homebased screening with IGRA. These included access, poor household ventilation, lack of privacy, stigma, sample storage and transportation to the central laboratory for testing.

## Access

*"Patients who have TB live in suburbs... to reach them you pass valleys and drainages, and you might actually need to park the car and get a motorcycle." (FGD_Hospital_4_Team_1_N01).*

*"...the roads were quite bad; they were not accessible." (KII_Hospital_3_N05)*

**Stigma.** Whereas index patients were welcoming and comfortable with the visiting study teams, some of their household contacts were concerned about the neighbors' perceptions as to why the study teams were visiting those particular homes in the villages. Thus, the study teams were invited to sit inside some poorly ventilated houses of the index patients houses. This was to prevent the neighbors from seeing what was going on which could have resulted into stigma.

*"The challenge that I found was stigma. ... the index TB patient was very inviting but when we reached the homes, the other parties, usually the wives, they had stigma.* (FGD_IGRA study team_ Mulago _1_NRH_1)

**Fear of injection.** Most people were fearing the injection. They thought it was taking off sputum. They were like, *"but for us we know TB is tested through sputum, and now you people are coming with injections..."* (KII_NURSING OFICER_Hoima_12)

**Poor ventilation.** Due to stigma, all activities had to be carried out inside the houses of the index TB patient. Majority of which had poor ventilation with no open widows. History of recurrent TB disease was noted in some of the homes.

*"... some of them the windows were completely sealed or were not opened, so we had to educate on infection control, but we had to enter those houses to do the activity." (FGD_Hospital_4_ Team_1_NRH_1).*

*"... we found about three homes which had contacts having TB recurrently; one particular home had about 3 people who had TB... (FGD_Hospital_ 4_ Team_1_NRH_1).*

**Sample storage and transportation.**    The test had to be transported to one central laboratory in the capital city. This limited time flexibility between sample collection, incubation, and analysis. Moreover, those processes had to be done under stringent conditions to ensure accurate results. The long distance increased the turn-around time & cost of the test. A dedicated team was required to ensure these timelines are met.

The participants also reported difficulty in transportation of samples from recipients' homes to the laboratory.

*"Transportation of samples with this recent experience; it appeared a bit difficult, but I know with time it will be improved." (KII_Hopsital_2_N 15)*

**Community response to the IGRA test.**    The healthcare workers found that the community was very accepting of IGRA testing. Community members who were contacts of confirmed cases were anxious to know if they were infected with TB while even those who were not contacts of the index case requested to be tested.

*"...the demand is really created because of the confirmed TB patients that are within the community. So, everyone is anxious to know their status (FGD_Hospital_4_Team_2 _N02.)*

*'...everybody was willing, and many other people wanted to take the test although they were not contacts. (KII_Hospital_3_N13).*

Even among child participants, IGRA uptake was very high. The community was receptive of needle pricks.

*I also want to comment on the phlebotomy, taking of blood. Frankly, I was impressed that—even the children, nobody cried. I also fear injections; okay I do not know how N04 did it... somehow even the children never cried. And I think majority of—index patients were very positive [about the IGRA test], and I think they did a good job in counselling the participants at home. Because the injection bit was received very well; even the children who were 5, 6, 7, they really did not cry.... (FGD_Hospital_4_Team _1_N01)*

*Preferred approach to LTBI screening using IGRA.* The acceptability of the test was due, in part, to the fact that a homebased screening approach was employed which such that no transport costs were incurred in the process of receiving care.

*"I think we can do with community [based contact tracing]—because in the community we deal with door to door, and in this case, we can handle many [more patients] than at the facility. Because at the facility, there are issues like transport costs that would prevent people from coming and in this case, we would get few, But, in the community, we go door to door where we can achieve more. (FGD_Hospital_4_Team_2 _N02).*

*So, I think it would be very hard for someone to just come at the facility to test for latent TB. Because if you are positive for latent TB, you don't have any symptoms; you are not coughing because it's not active. So, someone will come to test when they are active for TB, .... (KII_Laboratory 5_N10)*

*Willingness to pay for IGRA test.* The data shows that clients were not willing to pay for the test.

> *So, TB services are free—the TB test is free, the TB drugs are free, even the TB preventive therapy is free. Now, to make people pay for, for a test for uh, to test for hidden TB, for patients who already—you know TB is for poor people. . . (FGD_IGRA study team_Mulago _1_NRH_1) . . .*

## Discussion

Using a parallel convergent mixed methods design, we determined the prevalence of latent TB and health worker experiences in using IGRA home-based screening for LTBI. We found a LTBI prevalence of 32.7%. The risk factors associated with latent TB included being aged ≥ 45, being in formal employment or casual laborer, longer time spent with the index case, more intimate relationship with index case (parents or siblings) and sharing the same bedroom as the index case. The uptake and acceptability of the IGRA test among HHC of index TB patients was high at 99% and 95.7% respectively. Further, the test was viewed as useful by the health workers in detecting LTBI and bringing to light its true burden in our setting. Our study used a door-to-door approach, which provided the perspectives at the community level. It was found that the communities were receptive to the intervention. However, the challenges noted during IGRA implementation included difficult access to homes due to the poor state of roads in the slum dwellings, stigma, fear of injection, poor ventilation, challenges with sample storage and sample transportation, and delay in sample delivery.

The uptake and acceptability of the IGRA test in this study was generally high, however those who decline were mainly children aged 5–14 years. Refusal was uncommon (1%), similar to another study done amongst immigrants [13]. The main reason cited for refusal to take the test was pain from the needle prick.

The home-based approach to LTBI testing using IGRA could explain the high acceptability rates observed in our study. Similar home-based approaches in TB HHC investigation using other techniques like portable molecular diagnostics (portable GeneXpert-Instrument) [14] and home-based sputum collection [15] have showed that home-based approaches are convenient, trustworthy and help to overcome barriers to clinic-based testing like waiting time, distance and transportation costs.

The prevalence of LTBI determined in this study was lower than that reported by other studies in Uganda which reported prevalence that ranged from 51% to 65% [8, 16, 17]. Several reasons could explain the observed difference. Previous studies were carried out in urban or peri-urban setting which tend to have more crowding and poor ventilation which encourage transmission of TB infection. In addition, a study by Kizza et al, used TST rather than IGRA [8]. TST has a lower specificity than IGRA due to cross reactivity with BCG antigen and environmental non-tuberculous mycobacteria. Furthermore, our study population was a predominantly young population with the majority being between the (5–14) year age bracket compared to other studies where HHCs were older [8].

The factors associated with latent TB identified in our study were similar to those reported elsewhere. In India and China, LTBI was associated with increasing age and being in close contact to a case of tuberculosis [18, 19]. In addition, our study found that being employed as a casual labor was associated with a higher risk for LTBI positivity [19]. Older age increases the cumulative lifetime exposure to *Mycobacterium tuberculosis*, while being employed increases the risk for latent TB infection acquisition outside the household setting [19].

Similar to our study findings, other studies found that proximity of contact to a TB index case was associated with an LTBI positivity [20, 21]. In addition, our study showed that IGRA positivity was associated with increasing the time spent with the index TB patient which is

similar to what was found in India [22]. Presence of a BCG scar was not found to be statistically significant in our study, however, previous other studies have found BCG to be a protective factor against LTBl [23, 24]. This could be due to differences in the prevalence of TB between the study settings.

Despite the challenges experienced, IGRA based latent TB screening was well received by the community largely because it was free, and it was delivered at home. Free home-based latent TB screening overcame two of the major barriers to IGRA testing that includes transport costs and the need to pay for the test. A study carried out in Uganda to assess barriers to TPT uptake found that having to attend clinic refill visits and the need to pay for the service decreased participants willingness to initiate TPT [25].

Similar to findings elsewhere in the Netherlands [26] and Brazil [27], TB stigma was a major barrier to LTBI services. Increased knowledge and awareness of LTBI led to an increase in expressed stigma [27]. This was also the case in our setting were HHCs did not want the health worker teams to carry out any procedures from outside the house as they expressed fear of stigma from neighbors. To overcome these challenges, there is need to develop strategies that address stigma at the community level to help those affected to resist TB related stigma through counselling, creating TB support clubs, and community dialogues [28]. Strategies to decentralize laboratory testing capability will help address challenges of sample storage and transportation. Further, due to the current COVID-19 pandemic, only 63.2% received repeat IGRA testing at 9 months. This period was characterised by Index TB patients and their HHCs moving away from urban residences to rural areas for socio-economic reasons. Innovative patient-centred approaches need to be developed and evaluated as these will become increasingly relevant [1].

The study had several strengths and some limitations. We had regional representation from different parts of the country and so the findings are likely to be representative of different settings across the country. The study combined both qualitative and quantitative methods of data collection, which elucidated different perspectives of the study variables e.g., acceptability of the IGRA test, associated risk factors and barriers to implementation. Further it enabled triangulation of methods, data sources as well as researchers that enabled better understanding of the research questions.

One limitation of our study was the that the study population was heavily skewed towards children, given that children constituted the majority of HHCs in the study setting. The low sensitivity of IGRA in extremes of age [29] was mitigated by retesting at 9 months of follow up. In addition, this enabled identification of those who seroconvert later to be prioritized for LTBI treatment. Furthermore, our study had no age specific measures for acceptability, future studies should consider age-specific measures of acceptability to assess any differences in acceptability of the IGRA test among different age groups. Another study limitation is that perspectives in the qualitative research analysis were those of health personnel. More studies that explore the perspectives of TB household contacts need to be explored. Further, during the second study home visit, high rates of indeterminate IGRA results were reported as compared to first home visit. This may have been due to blood sample transportation delays due to political riots during pre-election campaigns.

Finally, the accrued sample size fell short of the estimated sample size due limited availability of test kits. However, our study's sample size was larger than for any prior study in Uganda and the study was spread across the country. Therefore, our study still gives the best estimate of the prevalence of LTBI in Uganda.

## Conclusion

Integrating home-based IGRA screening for LTBI into provision of TPT in routine care settings, resulted in high uptake and acceptability and was therefore feasible in a resource-limited

setting. Addressing challenges identified will be critical to scaling up IGRA based LTBI screening.

## Recommendations

1. Targeted IGRA testing for household contacts is acceptable and therefore national TB programs need to adopt IGRA based LTBI screening which has better specificity;

2. Home-based latent TB testing strategy should be incorporated into the national algorithm for latent TB management;

3. Laboratory capacity for IGRA testing needs to be decentralized to subnational or point of care level to overcome storage and transportation challenges;

4. There is need to evaluate the cost effectiveness of IGRA based LTBI testing and budget impact analysis in resource-limited settings to inform scale up.

## Supporting information

**S1 Data. A dataset of household contacts of index TB patients with data on socio-demographics and other detailed information on research study human subjects that participated in the IGRA study.**
(XLSX)

**S1 Text. Focus group discussion/key informant guide for healthcare worker study participants.**
(DOCX)

## Acknowledgments

The authors are indebted to TB index patients and members of their households who participated in the study. We thank the administration and health workers of the hospitals where this study was conducted for their invaluable contribution during the data collection process.

We acknowledge Adelline Twimukye (AT) a senior behavioural scientist from the infectious diseases institute for the technical guidance regarding all qualitative aspects of this study.

We appreciate the input and guidance of health authorities, the Ministry of Health-National TB and Leprosy Program (NTLP), Uganda.

## Author Contributions

**Conceptualization:** Jonathan Mayito, Abel Nkolo, Stavia Turyahabwe.

**Data curation:** Joseph Musaazi, Mabel Nakawooya, Didas Tugumisirize, Patrick Semanda.

**Formal analysis:** Simon Muchuro, Rita Makabayi-Mugabe, Joseph Musaazi, Stella Zawedde-Muyanja, Mabel Nakawooya.

**Funding acquisition:** Abel Nkolo, Stavia Turyahabwe.

**Investigation:** Simon Muchuro, Rita Makabayi-Mugabe, Jonathan Mayito, Stella Zawedde-Muyanja, Mabel Nakawooya, Didas Tugumisirize, Patrick Semanda, Steve Wandiga.

**Methodology:** Simon Muchuro, Joseph Musaazi, Jonathan Mayito, Stella Zawedde-Muyanja, Mabel Nakawooya, Didas Tugumisirize, Patrick Semanda, Steve Wandiga.

**Project administration:** Simon Muchuro, Stavia Turyahabwe.

**Software:** Joseph Musaazi.

**Supervision:** Simon Muchuro, Jonathan Mayito, Mabel Nakawooya, Didas Tugumisirize, Steve Wandiga, Susan Nabada-Ndidde, Abel Nkolo, Stavia Turyahabwe.

**Validation:** Simon Muchuro, Joseph Musaazi, Mabel Nakawooya, Didas Tugumisirize, Patrick Semanda.

**Writing – original draft:** Rita Makabayi-Mugabe.

**Writing – review & editing:** Simon Muchuro, Rita Makabayi-Mugabe, Joseph Musaazi, Jonathan Mayito, Stella Zawedde-Muyanja, Mabel Nakawooya, Didas Tugumisirize, Patrick Semanda, Steve Wandiga, Susan Nabada-Ndidde, Abel Nkolo, Stavia Turyahabwe.

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
