## [Decision Letter · Decision Letter 0]

26 Oct 2021

PGPH-D-21-00669

IGRA based latent TB screening for provision of tuberculosis preventive therapy is feasible in a TB high burden resource limited setting: a mixed methods study

Dear Dr. Makabayi-Mugabe,

Thank you for submitting your manuscript to PLOS Global Public Health. After careful consideration, we feel that it has merit but does not fully meet PLOS Global Public Health’s publication criteria as it currently stands. Therefore, we invite you to submit a revised version of the manuscript that addresses the points raised during the review process.

This is a significant topic for public health. LTBI screening and treatment is very important if the burden of TB is to be reduced and its relevance has been underestimated for decades in high-burden settings.

- Please include in the Abstract that IGRA were “home-based”. This is an important and distinctive contribution of this study that should be highlighted. The same comment for line 112 and 357

- As mentioned by a reviewer, it is not clear if any of the participants from the qualitative research analysis were TB contacts or there were all health personnel. This need to be clarified and acknowledged in the study limitations.

- Line 89: IGRAs are more specific than TST but I am not sure it has been demonstrated that IGRA have more “sensitivity” than TST; please revise this statement

- Methods: describe more clearly that TB contacts were sampled at home, and the conditions for blood samples transportation. Also please provide the version of the Quantiferon (Quantiferon TB-Gold or Quantiferon TB-Gold “In tube”?) and provider.

We look forward to receiving your revised manuscript.

Kind regards,

María Elvira Balcells, M.D., MSc

Academic Editor

Journal Requirements:

1. Please amend your detailed Financial Disclosure statement. This is published with the article, therefore should be completed in full sentences and contain the exact wording you wish to be published.

i). State the initials, alongside each funding source, of each author to receive each grant.

ii). State what role the funders took in the study. If the funders had no role in your study, please state: “The funders had no role in study design, data collection and analysis, decision to publish, or preparation of the manuscript.”

iii). If any authors received a salary from any of your funders, please state which authors and which funders.

2. In the online submission form, you indicated that "The datasets used and/or analyzed during the current study available from the corresponding author on request.". All PLOS journals now require all data underlying the findings described in their manuscript to be freely available to other researchers, either 1. In a public repository, 2. Within the manuscript itself, or 3. Uploaded as supplementary information.

3. Please provide separate figure files in .tif or .eps format only and remove any figures embedded in your manuscript file. Please ensure that all files are under our size limit of 20MB.  

Once you've converted your files to .tif or .eps, please also make sure that your figures meet our format requirements.

4. Please provide us with a direct link to the base layer of the map used in Fig1 and ensure this location is also included in the figure legend. 

Please note that, because all PLOS articles are published under a CC BY license (creativecommons.org/licenses/by/4.0/), we cannot publish proprietary maps such as Google Maps, Mapquest or other copyrighted maps. If your map was obtained from a copyrighted source please amend the figure so that the base map used is from an openly available source.

Please note that only the following CC BY licences are compatible with PLOS licence: CC BY 4.0, CC BY 2.0  and CC BY 3.0, meanwhile such licences as CC BY-ND 3.0 and others are not compatible due to additional restrictions. If you are unsure whether you can use a map or not, please do reach out and we will be able to help you. 

The following websites are good examples of where you can source open access or public domain maps:

Additional Editor Comments (if provided):

- Please change “GeneXpert” for Xpert MTB/RIF

- Please use abbreviations consistently (e.g. household contacts as HHC all over the text, same for LTBI)

- Please define “PR”

- Line 85: a comma is missing after ”supplies”

- Line 118: contacts were retested at 9 months, please revise as in Figure 2 indicates they were tested at 6 months. Was this also home-based testing?

-Line 135: there is an extra period

-Line 137: revise sentence writing

-Line 180: change the comma for a period

- Line 192-193: revise sentence writing

- Line 203: there is an extra space after the parenthesis

- Results: the proportion of indeterminate results for both visits should be provided in the text (in Fig 1 states 1% at baseline and 11% in the second visit which is quite high, any explanation?)

- Discussion line 392-303: revising the correct writing of "Mycobacterium tuberculosis"

Reviewers' comments:

Reviewer's Responses to Questions

**Comments to the Author**

1. Does this manuscript meet PLOS Global Public Health’s publication criteria? Is the manuscript technically sound, and do the data support the conclusions? The manuscript must describe methodologically and ethically rigorous research with conclusions that are appropriately drawn based on the data presented.

Reviewer #1: Yes

Reviewer #2: Yes

2. Has the statistical analysis been performed appropriately and rigorously?

Reviewer #1: Yes

Reviewer #2: Yes

3. Have the authors made all data underlying the findings in their manuscript fully available (please refer to the Data Availability Statement at the start of the manuscript PDF file)?

Reviewer #1: Yes

Reviewer #2: No

4. Is the manuscript presented in an intelligible fashion and written in standard English?

Reviewer #1: Yes

Reviewer #2: Yes

5. Review Comments to the Author

Reviewer #1: This study sought to determine the feasibility of incorporating testing for LTBI using IGRA into the provision of TPT in Uganda. The authors conducted a cross-sectional study at four tertiary referral hospitals that were geographically distinct. The total population of household contacts identified was 355. Feasibility was defined as rate of uptake of the test. This is an area of research that could benefit from a stronger evidence base.

Overall, this was a well written paper describing both quantitative and qualitative results regarding the acceptance and feasibility of home-based LTBI testing. Greater discussion around some of the key points made from the FGD and KII would strengthen the discussion. A more robust discussion about the limitations of the sample size stratified by age groups would also strengthen the paper. Detailed comments below.

1. Methods, data collection section, line 107, the authors state that data collection was carried out as part of household contact investigation. Given contact investigation is implemented in a variety of ways, adding a few sentences about this contact investigation activity would give a better picture of the intervention.

2. Methods, data collection section, line 109: the researchers are referred to as “we”. Who is we? A description of the study personnel and who is conducting the work at the household level would strengthen the methods.

3. Results section, line 184: please describe selection and randomization process. This is an important piece of the study design that has not been detailed.

4. Table 1: The high number of participants (39.2%) in the 5-14 year old age group surprised me here as I was reading and I wanted to go back and review what the outcome measures for acceptability were. How would someone in the 5-14 year age group differ from an adult in reporting acceptability? Were there any age-specific measures for acceptability? How defining “acceptability” was done in the younger age group could be better described in the methods.

5. Similar to the above comment, Table 2 shows that 14 of the 15 who reported poor acceptability were in the youngest age group. This needs to be addressed more clearly.

6. Table 2 shows that all of the 15 who reported poor IGRA acceptability were from the Hoima RRH health facility. Was this looked at more closely? Were there provider related factors influencing this finding?

7. Prevalence of LTBI. Only 63.2% received repeat IGRA testing at 9 months. Are there any hypotheses for this loss to follow up? Please address.

8. Table 3: Under “contact proximity” there is a variable “different house”. Given this is defined as household contact investigation, please define this category more clearly.

9. Line 33: “The acceptability of the test was due, in part, to the fact that a homebased screening approach was employed which such that no transport costs were incurred in the process of receiving care.” This is a very important point to raise in the results, along with the accompanying quote. This point is not expanded on in the discussion, but I would think is a key piece to the “feasibility” question. Please expand on this and frame in the literature in the discussion.

10. Discussion: Line 403: “… IGRA based latent TB screening was well received by the community largely because it was free, and it was delivered at home.” I think this point links directly to comment #9 and should be expanded on, including addressing the question about sustainability of this intervention.

Minor edits required:

Line 68: “falling” should read “fell” ad “in 2019” should be deleted as it is redundant.

Line 73: World Health Organization should be capitalized

Line 84: “its but” should be deleted

Line 244: Delete “constituted”

Line 390: replace “&” with “and”

Reviewer #2: Thank you for asking me to peer-review this manuscript. This is an important subject. LTBI screening and treatment is very important if the burden of TB is to be reduced but its importance is underestimated in the literature in high-burden settings.

This is a multicentre mixed methods study evaluating the uptake and acceptability of interferon-gamma release assay testing in a resource limited setting with a high incidence of tuberculosis. The study setting included four tertiary centres. The researchers recorded the uptake of IGRA-based LTBI screening among household contacts, aged over 5 years, of bacteriologically confirmed non-rifampicin resistant, non-multidrug resistant TB cases. IGRA screening was offered during a household visit. Active TB was excluded via a symptom screen. Data relating to duration and nature of contact of contacts with the case were collected also. The qualitative component of this research comprised of focus groups performed at the tertiary centre and interviews with individuals from the regional centres. It would seem the qualitative component only included healthcare providers.

I think the background requires more information for the reader to understand how the intervention (home-based IGRA screening) is different from typical practice. How are case contacts normally identified? Are contacts screened for LTBI as standard and if so how? Using TST? At the clinic or at home? By whom? In the introduction the authors state that the symptom screening tool is suboptimal for active TB exclusion. They go on to compare this with the TST. I found this confusing because in my mind it is drawing a comparison between a means of excluding active TB and a means of screening for LTBI. The authors go on to make valid comparisons between TST and IGRA use for LTBI screening. I think what is lacking from this comparison for the reader is how does uptake and acceptability of LTBI screening differ according to the test used in Uganda (or other high-incidence settings) among case contacts? Additionally, it should be noted that IGRA requires a laboratory and technician, trained phlebotomist and sample transport which may come at considerable cost compared with TST which may be more important in a resource limited setting. Again, I think some context as to what the current screening practice in Uganda is would help the reader determine what is new or different about this intervention in this setting.

I’m not sure feasibility is the correct term either. Feasibility, in my mind incorporates more than acceptability but also cost, practicalities such as transport (I believe IGRA samples must reach the laboratory and tested within 24-36 hours for the test to be valid) etc. Perhaps, acceptability is a better descriptor as to what this study aimed to measure.

The researchers say they randomly identified 86 index cases. I think it would be important to describe how these 86 cases were identified in the methods section where random selection is not mentioned. How many index TB patients were selected at random from which the 86 who agreed to participate arose? If the proportion of patients selected at random for enrolment who agreed to participate was low, then there is a risk of selection bias. Patients agreeing to participate probably are aware whether or not their household contacts would agree to LTBI screening, potentially biasing the uptake of IGRA testing. It would be good also to include a table of the TB patient characteristics to help the reader put into the context the population this study was performed in.

Please include a sentence as to why patients with drug resistant TB were excluded. This may be obvious to the researchers but wasn’t clear to me. A second IGRA was performed at 9 months, presumably to capture those who seroconvert at a later date but is there a scientific basis for the 9-month cut-off chosen which should be cited here?

The qualitative study methods appear appropriate. Perhaps list the questions in the main text or an appendix. Data collection and analysis was assisted by members of the research team. Is there a risk that researchers who were involved in the conduct of the quantitative research could have been biased towards inferring positive experiences of the intervention from the transcripts?

Table 1: I suggest changing the layout. Under employment, 60.8% have none. Is this because they are in education? If so, the employment heading should consider only those not in education.

Test uptake was exceptionally high, so too was reports of test experience as good/excellent. The authors say older age was associated with a bad phlebotomy experience. Should this not be younger age? Additionally, all those who had a negative phlebotomy experience were in one setting. This to me suggests it was related more to the setting. Was the phlebotomist here perhaps not as good at performing phlebotomy in children compared with the other hospitals? Could any inisghts be drawn on this from the qualitative analysis?

Regarding factors associated with a positive IGRA, multiple analyses are performed. These analyses appear to me to be exploratory and so it might be good to indicate in the methods section that all these analyses were not corrected for the multiplicity given the exploratory nature of the study. Additionally, depending on journal style, it may not be necessary or informative to include all P-values, particularly given the large number of analyses. Aside from this the factor identified as being associated with LTBI are logical and therefore believable and less likely to false positives. However, given the aim of the study was to determine the uptake and acceptability of the test is inclusion of these analyses even necessary? If so, I would suggest adding a sentence in the introduction to state this is a secondary aim.

The qualitative analysis was one third laboratory staff, would seem over represented to me and a limitation. Additionally, the qualitative analysis included none of the patients. The intervention seemed acceptable to the staff involved and they perceived it to be acceptable in general to the patients. It should be noted in the discussion the exclusion of patients was a weakness of the study design because some of the findings could not be explored further without their input e.g., the mentions of stigma by the healthcare workers and the high uptake and acceptability. Ultimately this study does not explain these further from the patients’ perspective. The healthcare workers seemed to think it was the avoidance of travel was the reason uptake was high but without asking the patients this cannot be said for sure. It may have been other factors related to the clinic or the homeplace or the behaviour of the staff (the staff had a positive experience, perhaps enthusiastic staff were more persuasive than they would have been in a clinic setting).

The discussion is acceptable.

Recommendations- I think it would be more correct to say “targeted IGRA screening for household contacts is acceptable” rather than “feasible” and the authors have gone on to say further evaluations are needed to determine its cost effectiveness (and they should add a budget impact analysis).

Some spelling and grammatical errors throughout.

To summarize, a well conducted mixed methods study on LTBI screening uptake and acceptability, an important subject for TB elimination globally. More information needed on what the current practice is regarding LTBI screening of contacts in Uganda (if any LTBI screening). More information on how the 86 cases were selected and their characteristics. Acceptability only evaluated from healthcare providers perspective which should be discussed as a major limitation.

6. PLOS authors have the option to publish the peer review history of their article (what does this mean?). If published, this will include your full peer review and any attached files.

**Do you want your identity to be public for this peer review?** For information about this choice, including consent withdrawal, please see our Privacy Policy.

Reviewer #1: No

Reviewer #2: **Yes: **James O'Connell

---

## [Editor Report · Decision Letter 1]

1 Feb 2022

PGPH-D-21-00669R1

IGRA based latent TB screening for provision of tuberculosis preventive therapy is feasible in a TB high burden resource limited setting: a mixed methods study

Dear Dr. Makabayi-Mugabe,

Thank you for submitting your manuscript to PLOS Global Public Health. After careful consideration, we feel that it has merit but does not fully meet PLOS Global Public Health’s publication criteria as it currently stands. Therefore, we invite you to submit a revised version of the manuscript that addresses the points raised during the review process.

We look forward to receiving your revised manuscript.

Kind regards,

María Elvira Balcells, M.D., MSc

Academic Editor

Journal Requirements:

1. We have noticed that you have uploaded supporting information but you have not included a list of legends. Please add a full list of legends for all supporting information files (including figures, table and data files) after the references list.

Additional Editor Comments (if provided):

Thanks authors for their great effort to improve the manuscript. The paper improved but there remain some issues that should be resolved.

ABSTRACT

• The current Abstract is too long, it must not exceed 300 words: https://journals.plos.org/globalpublichealth/s/submission-guidelines

• line 32: “that would require TPT was unknown”: change to ”that would require TPT is unknown”

• line 38: “without signs or symptoms of TB”: change to ”without signs of symptoms of active TB”

• line 52 (and also in line 269): “older age >45”: “>45 years old”…

• line 59: “for sample storage”: change to ”for blood sample storage”…

• line 61: ”IGRA based”: change to ”Home-based IGRA screening”…

• line 86 (and again at line 119): “PBCs” Is undefined (please avoid too many abbreviations)

• Line 51-52: IGRA negative results testing positive at 9 months are 4.3%, but in Results section of the manuscript, the same proportion is reported as 3.4% (line 265). Which is the correct?

INTRODUCTION

• lines 81-89: Please reformulate, as the new paragraph is a little bit confusing regarding what is recommended to screen active TB versus latent TB. Furthermore, the sentence “Although TST has better sensitivity and specificity than the WHO symptom screen,” is not correct as the symptom screening algorithm is purposed to find active TB, and TST is for finding LTBI. Additionally, what is currently done as standard of care for HHC in Uganda is still unclear in the Introduction.

• IGRA: there are more than one IGRA available on the market, so I suggest changing to plural (IGRAs) when referring to any IGRA.

METHODS

• A sample size calculation for survey is now provided in the corrected version of the manuscript . However , the authors subtract 10% accounting for those HHC that may be found later having active TB. I think I would do the opposite (adding 10% to account for those that will have to be excluded). Unless you assume that the n=385 already includes only those fulfilling inclusion and exclusion criteria, in which case you may just eliminate the sentence line 113-114. Also, revise spelling and clarity of the sentences (e.g “formular” in line 110), please refer to standard way of reporting sample sizes calculations.

• Line 119: “Of these 299 patients...”: change to: “Of these, 299 patients..”

• Line 127: “One index TB patients declined” : removed the “s” from patients

• Line 133-134: It is important to describe how the 86 index cases were randomly selected (what was the proportion for all TB cases found in the same period Feb to Dec 2020)

RESULTS

• As previously raised by one of the reviewers, children under 18 should not be included in the “unemployed category”. Employment status or type should rather include only those not on education (Table 1)

• Line 287-292 and Table 4. It should include which was the proportion of all index cases included in this study for which information was available (53 out of 86)

DISCUSSION

• Line 448: “the prevalence determined in this study”: change to ”the prevalence of latent TB infection (or LTBI) determined in this study”

• Line 450-451: “studies were carried out urban or pre-urban setting”: change to ”studies were carried out in urban or pre-urban setting”

• “Further,...”: change to: “Furthermore,...”

• Line 507-508: “high rates of indeterminate results were reported”: change to “high rates of indeterminate IGRA results were reported”

• Line 509-511: the sentence comment on age-specific measures to increase acceptability could be introduced by : Finally,… (as it is not related to the previous limitation in the text), or maybe moved up in the same paragraph (e.g. at the end of line 502).

CONCLUSION:

• Line 514: “IGRA based screening had a high uptake and acceptability and therefore feasible in …”: change to ”Home-based IGRA screening for latent tuberculosis infection had a high uptake and acceptability and therefore is feasible in …”

ALL OVER THE TEXT

• When “interferon gamma release assay”: change to “interferon-gamma release assay”

• “IGRA QuantiFERON-TB Gold Plus test” is a little long. I suggest to abbreviate as QFT or QFT-G. And it is not necessary to put both together, just IGRA or just QFT depending on context (but not “IGRA QuantiFERON-TB Gold Plus”

• Revise once again the consistency of all abbreviations (LTBI, TB, PTB, etc) all over the text

• “IGRA based”: change to “IGRA-based”

• “homebased” or “home base”: change to ”home-based”
---

## [Editor Report · Decision Letter 2]

24 May 2022

PGPH-D-21-00669R2

Integrating Interferon Gamma Release Assay testing into provision of tuberculosis preventive therapy is feasible in a TB high burden resource limited setting: a mixed methods study

Dear Dr. Makabayi-Mugabe,

Thank you for submitting your manuscript to PLOS Global Public Health. After careful consideration, we feel that it has merit but does not fully meet PLOS Global Public Health’s publication criteria as it currently stands. Therefore, we invite you to submit a revised version of the manuscript that addresses the points raised during the review process.

We look forward to receiving your revised manuscript.

Kind regards,

María Elvira Balcells, M.D., MSc

Academic Editor

Journal Requirements:

1. Please update your Competing Interests statement. If you have no competing interests to declare, please state: “The authors have declared that no competing interests exist.”

2. Please provide an Author Summary. This should appear in your manuscript between the Abstract (if applicable) and the Introduction, and should be 150–200 words long. The aim should be to make your findings accessible to a wide audience that includes both scientists and non-scientists. Sample summaries can be found on our website under Submission Guidelines: https://journals.plos.org/globalpublichealth/s/submission-guidelines#loc-parts-of-a-submission

Alternative link: http://journals.plos.org/ploscompbiol/s/submission-guidelines#loc-author-summary

Additional Editor Comments (if provided):

1. Title:

Replace the abbreviation “TB” with “tuberculosis”; “interferon gamma” with “interferon-gamma” (revise this in all the manuscript); “resource limited” with “resource-limited”

2. Abstract:

• Line 34: I suggest changing “pulmonary bacteriologically confirmed tuberculosis” to “bacteriologically confirmed pulmonary tuberculosis” (the same for Table 4)

• Please restore the measures of association (e.g. PR and 95%CI) for factors significantly associated with LTBI in the Abstract

• Also, why other factors significantly associated with LTBI such as the type of work were removed from the abstract?

• The authors removed the word “home-based” from Abstract conclusion, I suggest re-including it, as this is an important novelty of this study

3. Introduction:

• Review this sentence line 71-72: “Although immunological tests e.g.,the Tuberculin Skin Test (TST) have better sensitivity and specificity than the WHO symptom screen,…” (“as” is lacking before “the TST”?)

4. Methods:

• Line 86-87: Please provide the name of participating hospitals centers

• Line 107 states “Drug resistant TB patients were not eligible for TPT…” Please correct to “HHCs of drug-resistant TB patients were not eligible for TPT…”

• Sample size: 106 index cases for 424 HHCs were required. The authors state that only 86 index cases and 352 HHCs were enrolled “due to limited availability of test kits”. Therefore, revise line 115 as the sentence “Consequently, 86 index TB case’s homes were visited…” does not make it very clear at which point or which of the four participating hospitals did not complete the sample size. Was the sampling proportionate to size maintained?

5. Results

• Spelling revise and correct to: Line 223 “follow-up”; line 229 “the” majority; line 241 “p value =0.02”); line 243 “unadjusted”

• Table 4: format the table results as previous tables; e.g. the percentages are included in the same column as the absolute numbers, e.g. 14(26.4%). The dashs for age intervals should be at the middle, e.g 15-17 and not 15_17, >45 and not 45+, etc. Line 281(Table footnote) revise the sentence for clarity.

• Line 288-289: “we carried out two focus group discussions (FGDs) each with five participants and 14 key informant interviews (KII) giving a total of 24 participants” (5+14=19, not 24…please revise). Also, importantly, please change “participant” for “health care worker participants in this study “ (or something alike) to not confuse readers with HHCs participants.

6. Discussion

• Line 437 “Tuberculin Skin Test” was already defined as “TST” in previous paragraphs.

• Line 447 introducing M.tb abbreviation seems unnecessary here

• Line 448 change “latent TB infection outside the household…” for “latent TB infection acquisition outside the household…”

7. Funding: line 545 states “None of the authors received a salary from the authors.” Should it be “…. salary from the funders”?
---

## [Editor Report · Decision Letter 3]

1 Jun 2022

Integrating Interferon-Gamma Release Assay testing into provision of tuberculosis preventive therapy is feasible in a tuberculosis high burden resource-limited setting: a mixed methods study

PGPH-D-21-00669R3

Dear Dr Makabayi-Mugabe,

We are pleased to inform you that your manuscript 'Integrating Interferon-Gamma Release Assay testing into provision of tuberculosis preventive therapy is feasible in a tuberculosis high burden resource-limited setting: a mixed methods study' has been provisionally accepted for publication in PLOS Global Public Health.

Best regards,

María Elvira Balcells, M.D., MSc

Academic Editor

All minor corrections were adequately addressed.